# Nuclear Factor-κB Decoy Oligodeoxynucleotide Attenuates Cartilage Resorption In Vitro

**DOI:** 10.3390/bioengineering11010046

**Published:** 2024-01-01

**Authors:** Hitoshi Nemoto, Daisuke Sakai, Deborah Watson, Koichi Masuda

**Affiliations:** 1Department of Otolaryngology-Head and Neck Surgery, School of Medicine, University of California, La Jolla, San Diego, CA 92093, USA; 2Department of Plastic Surgery, School of Medicine, Tokai University, Isehara 259-1193, Kanagawa, Japan; 3Department of Orthopaedic Surgery, School of Medicine, University of California, La Jolla, San Diego, CA 92093, USA; daisakai@is.icc.u-tokai.ac.jp (D.S.); komasuda@health.ucsd.edu (K.M.); 4Department of Orthopaedic Surgery, School of Medicine, Tokai University, Isehara 259-1193, Kanagawa, Japan

**Keywords:** nuclear factor-κb, decoy oligodeoxynucleotide, cartilage, resorption

## Abstract

Background: Cartilage harvest and transplantation is a common surgery using costal, auricular, and septal cartilage for craniofacial reconstruction. However, absorption and warping of the cartilage grafts can occur due to inflammatory factors associated with wound healing. Transcription factor nuclear factor-κB (NF-κB) is activated by the various stimulation such as interleukin-1 (IL-1), and plays a central role in the transactivation of this inflammatory cytokine gene. Inhibition of NF-κB may have anti-inflammatory effects. The aim of this study was to explore the potential of an NF-κB decoy oligodeoxynucleotide (Decoy) as a chondroprotective agent. Materials and Methods: Safe and efficacious concentrations of Decoy were assessed using rabbit nasal septal chondrocytes (rNSChs) and assays for cytotoxicity, proteoglycan (PG) synthesis, and PG turnover were carried out. The efficacious concentration of Decoy determined from the rNSChs was then applied to human nasal septal cartilage (hNSC) in vitro and analyzed for PG turnover, the levels of inflammatory markers, and catabolic enzymes in explant-conditioned culture medium. Results: Over the range of Decoy conditions and concentrations, no inhibition of PG synthesis or cytotoxicity was observed. Decoy at 10 μM effectively inhibited PG degradation in the hNSC explant, prolonging PG half-life by 63% and decreasing matrix metalloprotease 3 (MMP-3) by 70.7% (*p* = 0.027). Conclusions: Decoy may be considered a novel chondroprotective therapeutic agent in cartilage transplantation due to its ability to inhibit cartilage degradation due to inflammation cytokines.

## 1. Background

Cartilage grafting is a common surgical technique in plastic and reconstructive surgery. Tissue sources include cartilage from the rib, nasal septum, and ear for nasal reconstruction, rhinoplasty, microtia repair, and eyelid reconstruction. However, grafts of cartilage have been associated with post-graft resorption and warping [1,2]. Clinically, NSC is an excellent donor source material [3], but resorption rates in the range of 12 to 50 percent have been reported [4,5]. Resorption or warping of the grafted cartilage can be clinically problematic. NSC is also a promising cell source for tissue-engineered cartilage grafts (e.g., small amounts of cartilage are harvested, cells and extracellular matrix (ECM) are amplified in vitro, and cartilage tissue is constructed and then transplanted into the body where it is needed.) [6], and methods and agents to prevent resorption and warping are required for transplantation of both as-is and tissue-engineered cartilage. Factors that tend to affect graft absorption and warping include: high-tensile forces [4], compressive forces [7], insufficient nutrition and cell source within the recipient area [3], and the inflammatory process associated with wound healing [8].

It is well known that wound healing involves three phases: inflammation, proliferation, and remodeling [9]. The inflammatory phase is triggered by many sources of tissue injury, including surgical procedures. Leukocytes are activated by the release of various cytokines, such as interleukin-1 (IL-1), interleukin-6 (IL-6), and tumor necrosis factor-α (TNF), during this initial phase [10]. This is followed by fibroblast proliferation, angiogenesis, and keratinocyte migration. The release of cytokines can cause net catabolic remodeling of the cartilage graft creating an imbalance in graft matrix degradation relative to matrix synthesis [8]. In recognizing this, Haisch et al. [11] showed that encapsulation of the graft with a polyelectrolyte complex membrane offers a protective barrier for the cartilage implant against an inflammatory reaction. This suggests that ways of locally inhibiting cartilage catabolism may be beneficial.

The transcription factor nuclear factor-κB (NF-κB) plays a central role in inflammation and the immune response. There are two types of NF-κB activity pathways (classical and non-classical): the classical pathway is activated by proinflammatory cytokines such as TNF, IL-1, IL-6, chemokines, and anti-microbial proteins cytokines, while the non-classical pathway is activated by CD40L and B cell activating factor (BAFF) [12]. NF-κB was originally discovered as a lymphoid-specific protein that binds to the decameric oligodeoxynucleotide (ODN) GGGACTTCC present in the intronic enhancer element of the immunoglobulin κ light chain (Iκ) gene [13]. In the classical pathway, NF-κB is normally present in the cytoplasm in an inactivated state by binding inhibitor proteins including IκB [14,15]. IκB is phosphorylated and undergoes degradation through multi-subunit IκB kinase (IKK) activation [16]. Free NF-κB rapidly enters the nucleus and transactivates target genes such as TNF, IL-1, and IL-6 [14,17]. Theoretically, inhibition of NF-κB may provide an anti-inflammatory effect.

The possibility of a therapeutic agent which uses dsDNA as a “decoy” with a high affinity to targeted transcription factors was first described by Sullenger et al. and Bielinska et al. [18,19]. Although antisense ODN has the ability to block target genes, decoy ODN is more effective in blocking constitutively expressed factors, as well as multiple transcription factors that bind to the same cis element [20]. NF-κB decoy oligodeoxynucleotide (Decoy) is a synthetic double-stranded deoxyribonucleic acid (dsDNA) containing the classical NFkB-P65 binding site (GGGATTTCCC) which has a high affinity to activated NF-κB. In this regard, activated NF-κB binds to Decoy instead of the DNA enhancer elements resulting in inhibition of NF-κB (Figure 1) [13,14,20,21]. The effect of Decoy on the reduction in the extent of myocardial infarction following reperfusion in a rat model via inhibition of IL-6 and intercellular cell adhesion molecule (ICAM) was first demonstrated in vivo by Morishita et al. [22]. Consequently, it was realized that Decoy was not only effective against ischemia and reperfusion injury, but it also had anti-tumor and anti-inflammatory effects [17,23,24,25,26,27,28,29].

The effectiveness of Decoy has been shown in a variety of previous studies including pulmonary metastasis by murine osteosarcoma [23], acute renal failure in rats [24], liver grafts in rats [25], rheumatoid arthritis in human synovial cells [26], sun damage of skin in mice [27], ischemic injury in a rat epigastric flap model [17], inflammatory bowel disease in rats [28], and intervertebral disc (IVD) deterioration in rabbits [21]. In anticipation of this anti-inflammatory effect, a phase 1b clinical trial of Decoy for chronic discogenic lumbar back pain was initiated in the United States in February 2018 [30]. The results of this phase 1b clinical trial showed no significant adverse events and confirmed improvement in low back pain for 12 months after administration, and a phase 2 clinical trial was initiated in Japan in October 2023 [31]. A chimeric decoy ODN, which inhibits both NF-κB and the allergic mediator signal transducer and activator of transcription 6 (STAT6), was also developed and used in a mouse model for asthma. It has been reported that chimeric decoy ODN had greater anti-inflammatory effects than decoy ODN with NF-κB alone in the mouse asthma model [32]. Although its efficacy has been confirmed for a variety of applications, it has not yet been explored in the nasal septum cartilage.

The level of the load-bearing ECM component, proteoglycan (PG), reflects a cytokine-regulated dynamic balance between PG synthesis by indwelling chondrocytes and PG degradation in articular cartilage [33]. In articular cartilage explants, serum stimulates PG synthesis [34] and inhibits PG degradation [35]. Within serum, insulin-like growth factors normally mediate the stimulation of PG synthesis and inhibition of PG degradation [36], whereas the catabolic factor from synovium [37], found to be IL-1 [38], tips the PG balance to a catabolic state, inhibiting PG synthesis and stimulating PG degradation [39]. It has also been shown that IL-1β regulates the expression of catabolic factors in mouse chondrocytes [40]. IL-1 induces formation of nitric oxide (NO) and matrix-depleting enzymes including matrix metalloproteinase-3 (MMP-3). MMP-3 causes PG depletion and cartilage softening [41], while prolonged IL-1 treatment leads to collagen network damage and loss of cartilage integrity [42]. Anti-cytokine protein agents can prevent experimental arthritis following traumatic injury [43]. Thus, the aim of this study was to assess the potential of Decoy as a chondroprotective agent, inhibiting the degradation of PG in hNSC tissue, associated with NO and MMP-3 formation. Pretreatment of hNSC tissues with Decoy may prevent early resorption of transplanted tissues and improve the efficacy of cartilage transplantation.

## 2. Materials and Methods

The study design had two parts. In the first part, the evaluation of safety and determination of an efficacious concentration of Decoy was performed using rabbit nasal septal chondrocytes (rNSChs). In the second part, this determined concentration of Decoy was used for human nasal septal cartilage (hNSC). PG turnover assay and the determination of protein levels of inflammation makers were completed. Decoy 5′-CCTTGAAGGGATTTCCCTCC-3′ and 3′-GGAACTTCCCTAAAGGGAGG-5′ was provided by AnGes MG (Osaka, Japan). Decoys used in this study were phosphonothioate-modified. Decoy is effective in binding the NF-κB transcription factor [22,44,45].

### 2.1. rNSCh Cell Isolation and Culture in Alginate Beads

After euthanasia, cartilage from the nasal septum was aseptically harvested from 16 New Zealand White rabbits weighing 3.5–4.5 kg (Institutional Animal Care and Use Committee approval #S08258). rNSChs were isolated by sequential enzyme digestion for 1 h at 37 °C with 0.2% pronase (53702, Millipore, Billerica, MA, USA) in Dulbecco’s Modified Eagle Medium and Ham’s F12 medium (10-090-CV, DMEM/F-12; Corning Inc. Corning, NY, USA) [46]. Afterwards, the tissues were digested overnight with 0.025% collagenase-P (11213865001, Roche Diagnostics, Indianapolis, IN, USA) in DMEM/F12 medium with 5% fetal bovine serum (FBS; FB-03, Omega Scientific, Tarzana, CA, USA) [46]. After overnight digestion, the cells were washed with DMEM/F-12 medium, filtered through 70 μm nylon mesh (22-363-548, Fisher Scientific, Pittsburgh, PA, USA) and cell numbers were counted. The cells were then seeded in a T175 flask (CLS431080, Corning Inc.) at 10,000 cells/cm^2^ and cultured for expansion with culture medium (DMEM/F-12 medium containing 10% FBS, 25 μg/mL ascorbic acid (50-81-7, Sigma-Aldrich, St. Louis, MO, USA), 360 μg/mL L-glutamine (25-005-CI, Corning Inc.), antibiotic–antimycotic solution (100 U/mL penicillin G, 100 μg/mL streptomycin sulfate, and 0.25 μg/mL amphotericin B) (30-004-CI, Corning Inc.) and 50 μg/mL gentamicin (15750060, Thermo Fisher Scientific, Waltham, MA, USA)). The cells were released from monolayer with 0.25% trypsin ethylenediaminetetraacetic acid (EDTA; 17892, Thermo Fisher Scientific) at confluence. The isolated cells were resuspended in a 1.2% low-viscosity sterile alginate (Keltone LV; Kelco, Chicago, IL, USA) solution at a concentration of 2 million cells/mL. Polymerization and formation of the alginate beads was accomplished by injecting the cell–alginate mixture into a 102 mM CaCl_2_ solution in a dropwise fashion through a 22-gauge needle [46]. After 10 min, the newly formed beads (approximately 20,000 cells/bead) were washed three times with a sterile 0.9% NaCl solution followed by two washings with DMEM/F-12. The cells in the beads (9 beads per well) were cultured in a 24-well culture plate (CLS3527, Corning Inc.) with complete medium (DMEM/F-12 medium containing 20% FBS, 25 μg/mL ascorbic acid, 360 μg/mL L-glutamine, antibiotic–antimycotic solution, and 50 μg/mL gentamicin). The cultures were maintained at 37 °C in a humidified atmosphere of 5% CO_2_. The medium was changed 3 times per week.

### 2.2. Cytotoxicity Analysis of Decoy

As an indicator of cell death by plasma membrane damage, the release of lactate dehydrogenase (LDH) within the culture medium was measured using a Cytotoxicity Detection Kit (11644793001, Merck, Darmstadt, Germany). Following the culture of rNSCh-encapsulated alginate beads for 7 days, Decoy (0 and 10 μM) was transfected to the chondrocytes within the beads. The culture medium was collected after 44 h of incubation and assayed for LDH activity according to the manufacturer’s instructions. The in vitro release of LDH from the cells provides an accurate measure of dead cells [47]. Culture medium was collected from the chondrocytes within the beads, with 0.9% normal saline serving as the control and 0.1% Tween 20 (9005-64-5, Merck) serving as the positive control.

### 2.3. rNSCh PG Synthesis Assay

On day 7 of cell culture in the alginate beads, the cells were incubated in complete medium in the presence of Decoy and recombinant human IL-1β (IL-1; SIL1B, Thermo Fisher Scientific) (control, Decoy 10 μM, IL-1 5 ng/mL, IL-1 5 ng/mL + Decoy 1 μM, and IL-1 5 ng/mL + Decoy 10 μM). During the last 4 h of culture time, the medium was changed, in each case, to complete medium with the treatment containing [^35^S]-sulfate (PerkinElmer, Waltham, MA, USA) at a concentration of 20 µCi/mL. After removing the medium, the beads were dissolved with 55 mM sodium citrate, 30 mM EDTA, and 150 mM sodium chloride and the two compartments (cell-associate matrix (CM) and further removed matrix (FRM)) were separated by centrifugation at 100× *g* for 10 min. Each fraction was digested with papain (76216, Merck) and incorporation of [^35^S]-sulfate was determined using a rapid filtration assay after precipitation of the glycosaminoglycans (GAGs) with alcian blue [48]. Rates of [^35^S] incorporation were calculated and expressed as CPM [^35^S]-incorporated/9 beads.

### 2.4. rNSCh PG Turnover Assay

On day 7 of cell culture in the alginate beads, the cells were incubated in complete medium containing [^35^S]-sulfate at a concentration of 20 µCi/mL. Following 16 h incubation, the beads were washed 3 times with complete media containing a total of 1.5 mM sulfate in order to removed unincorporated [^35^S]-sulfate. Five groups were created with different dosing concentrations of Decoy and IL-1 (0:0 (control), 10:0, 0:5, 1:5, and 10 μM:5 ng/mL). The medium was changed to complete medium with Decoy. Following 4 h of transfection of Decoy, IL-1 was added. The medium containing Decoy and IL-1 was changed and collected every 24 h. On day 5 from the start of treatment, the beads were dissolved and separated to CM and FRM and then digested with papain [^35^S]-sulfate in the collected media and digested CM and FRM were determined using an alcian-blue rapid filtration assay.

Half-life of PG was calculated using the single exponential decay method [49].

### 2.5. hNSC Harvest and Tissue Culture

hNSC specimens were collected from four patients (30-year-old female, 48-year-old male, 49-year-old male, and 58-year-old female) undergoing routine septoplasty and septorhinoplasty at our institution (institutional review board approval, project #130631X). Specimens were transported in sterile saline to the lab. Perichondrium was meticulously removed from cartilage. Within 24 h of explant, each cartilage specimen was weighed and minced into 10–50 mg fragments for tissue culture. The diced cartilage was cultured with complete medium throughout the study. The cultures were maintained at 37 °C in a humidified atmosphere of 5% CO_2_.

### 2.6. Efficiency of Transfection of Decoy

To study transfection efficiency of Decoy in the tissue culture system, fluorescein isothiocyanate (FITC)-labeled Decoy (gifted by AnGes MG) was used. A fragment of whole hNSC was divided in half; one half was cultured for 4 h in complete media containing 1 μM of FITC-labeled Decoy. Following washing and incubation for 1h with complete media, the tissue was examined by a confocal spectral laser scanning microscope (Leica Microsystems GmbH, Wetzlar, Germany). To further confirm the cytotoxicity of decoy, the live–dead assay of the second half of tissue was performed using a LIVE/DEAD Viability/Cytotoxicity Kit for mammalian cells (L3224, Thermo Fisher Scientific) according to manufacturer’s guidelines.

### 2.7. hNSC PG Turnover Assay

Following the method described above, after 24 h of culture, cartilage was incubated in complete medium containing [^35^S]-sulfate at a concentration of 20 µCi/mL. Following 16 h incubation, the tissue was washed. The medium was changed to complete medium with Decoy and IL-1 (0:0, 10:0, 0:5, and 10 μM:5 ng/mL). The medium containing Decoy and IL-1 was changed and collected every 24 h. On day 5 from the start of treatment, the tissue was collected and digested with papain. [^35^S]-sulfate in the collected media and digested tissue was determined using an alcian-blue rapid filtration assay. Half-life of PG was calculated based on fitting [^35^S] loss to an exponential decay.

### 2.8. Enzyme-Linked Immunosorbent Assay (ELISA) Matrix Metalloprotease 3 (MMP-3), TNF, and IL-6

After 2 days of treatment with IL-1 and Decoy in the tissue culture, the media were assayed by ELISA using the Human MMP-3 DuoSet (DY513, R&D systems Inc., Minneapolis, MN, USA), the Human TNF High Sensitivity ELISA kit (BMS223HS, Thermo Fisher Scientific), and the IL-6 ELISA Ready-SET-GO kit (88-7066-76, Fisher Scientific) according to manufacturer’s guidelines. All samples were run in duplicate and the resulting quantities were averaged.

### 2.9. Measurement of Nitric Oxide (NO)

After 2 days of treatment with IL-1 and Decoy in the tissue culture, total nitrate and nitrite concentration in the media was also assayed using a Nitrate/Nitrite Colorimetric Assay Kit (780001, Cayman Chemical Company, Ann Arbor, MI, USA) according to the manufacturer’s instructions. All samples were run in duplicate and the resulting quantities were averaged.

### 2.10. Statistical Analysis

Any assay which showed data to be out of range by the average ± 2 standard deviation (SD) was eliminated. The Levene test was used to assess the equality of variances. An unpaired *t*-test was used to analyze the difference between the two groups. An analysis of variance (ANOVA) was performed with Fisher’s LSD as a post hoc test. If equal variances were rejected, the Games–Howell method was used as a post hoc test. In multi-way ANOVA, Mauchly’s sphericity test was performed, and if sphericity was rejected, the degrees of freedom were corrected with ε calculated by the Greenhouse–Geisser method (SPSS ver. 26, IBM, Chicago, IL, USA). All results are expressed as the mean ± standard error of mean. Statistical significance was taken at *p* ± 0.05.

## 3. Results

### 3.1. Decoy Inhibiting the Release of LDH

Positive controls were set at 100%. Decoy did not affect cultured rNSChs in terms of cell viability or PG synthesis. Decoy did not show a cytotoxic effect based on the LDH assay (Decoy of 16.5 ± 3.4% vs. control of 19.8 ± 2.0%, *p* = 0.44, Figure 2).

### 3.2. Decoy Not Affecting PG Synthesis

For the PG synthesis assay, compared with the control, IL-1 administration had a >30% reduction (*p* < 0.0001), whereas the 10 μM Decoy group did not inhibit PG synthesis (*p* = 0.263). Compared with IL-1, Decoy did not inhibit the attenuation of PG synthesis by IL-1 (IL-1 + 1 μM Decoy, *p* = 0.430; IL-1 + 10 μM Decoy, *p* = 0.132) (Figure 3).

### 3.3. Inhibiting of PG Degradation in rNSCh by 10 μM Decoy

Compared with the controls, the 10 μM decoy group showed significant inhibition of PG degradation in the PG turnover assay (10 μM decoy group on day 5: 96.45 ± 0.53% and control on day 5: 93.69 ± 1.1%, *p* < 0.0001). Although PG retention in the IL-1 group was lower than in the control group, there was no significant difference (IL-1 on day 5: 92.45 ± 0.85%, *p* = 0.821). In the presence of IL-1, Decoy at 1 μM concentration did not inhibit PG degradation (IL-1 + 1 μM Decoy on day 5: 93.33 ± 0.94%, *p* = 0.394), while the 10 µM concentration of Decoy inhibited degradation of PG significantly (IL-1 + 10 μM Decoy on day 5: 96.1 ± 0.8%, *p* < 0.0001) (Figure 4).

### 3.4. Decoy Transfected into Viable Cells

The transfection of FITC-labeled Decoy to the chondrocytes was identifiable 4 h after initial administration (Figure 5a). The live–dead assay demonstrated that most of the cells in the tissue were live cells (Figure 5b,c). From these images, it was inferred that Decoy was incorporated into viable cells.

### 3.5. Inhibiting of PG Degradation in hNSC Tissue Culture by 10 μM Decoy

PG turnover was affected by individual patient differences (*p* < 0.001), treatment conditions (*p* < 0.001), and duration of culture (*p* < 0.001), with an interactive effect (*p* = 0.001, Figure 6). Decoy significantly inhibited degradation of PG with or without IL-1. On day 5 of culture, the retention of PG for the control group (73.6 ± 7.3%) was increased by Decoy (to 83.6 ± 2.5%, *p* < 0.01), and for the IL-1 (73.9 ± 7.1%), it was also increased for IL-1 + Decoy (82.5 ± 2.5%, *p* < 0.01). The half-life of PG was prolonged 63% by addition of Decoy under both conditions (control: 11.5 days, Decoy: 19.3 days, IL-1: 11.5 days, and IL-1 + Decoy: 18.7 days). There was no detectable difference between the control group and the IL-1 group (*p* = 1.00).

### 3.6. Decoy Inhibiting MMP-3 Production (ELISA and Nitrate Assay)

MMP-3 was decreased by more than 70% with administration of Decoy compared with the controls (*p* < 0.05). IL-1 treatment increased MMP-3 to 301.3% of controls (*p* < 0.001). In the presence of IL-1, MMP-3 was also significantly reduced by 51.1% with Decoy administration (*p* < 0.001). (Figure 7). Decoy did not inhibit the other inflammation makers, TNF, IL-6, or NO (Figure 8a–c).

## 4. Discussion

ODN (including Decoy) is mainly present in the ECM for 2 h after administration and then migrates into the cell in a time-dependent manner [50]. All Decoy that enters the cell enters the nucleus [21]. The mechanism of action of Decoy in the nucleus was previously described but its duration within the nucleus is not well known. In general, ODN is readily degraded to DNase both in vitro and in vivo [50,51]. Within 5 min of intravenous administration, 30–40% of all ODN is missing at least one nucleotide [50]; however, this depends on whether it is administered intravenously or locally. Modifications to ODN may also have an impact, for example, on hydrogen bonding and phosphonothioate [21]. Interestingly, Decoy has been found to be present in joints after 28 days of administration [52]. In addition, when Decoy was injected topically into rabbit IVD, the corresponding disappearance/distribution half-lives were 11.9 and 618 h, respectively, and Decoy was still present after 4 weeks, as was observed in joints [21]. Conversely, when plasmids containing decoys were administered intravenously to mice, the time window was only 2 h [53].

The elasticity of cartilage enhances its kinetic function in the joints and respiratory function in the nose. The physical properties of cartilage are determined by its specialized ECM and moisture content. The composition of the ECM of NSC is (per gram of wet cartilage) 7.39 μg collagen (mainly collagen II) and 1.71 μg sulfated glycosaminoglycan (PG) [54]. Although the percentage of PG is low, the strong water retention and elasticity are due to hyaluronic acid and PG (mainly aggrecan). The study of PG dynamics usually refers to the maintenance or loss of cartilage itself. Methods for quantifying PG include dye-binding methods such as dimethyl methylene blue (DMB), Western blotting, and chromatography (with some limitations) [48,55]. These methods detect the content of PG, by reflecting the results of catabolic and anabolic activities. Our study utilized a radioactive pre-labeled method followed by a chase (the pulse–chase method) which is a gold standard used to identify the rate of PG degradation, separately from its anabolic activity [56]. This method has advantages over gene expression analysis, PG content detection, and Western-blot methods as it is able to identify the net catabolic status of tissues [24,25,26,27,28,29,30], although the specific enzymes involved cannot be revealed without the newly cleaved epitope analysis.

Major enzymes that degrade the ECM of cartilage, such as collagenase, originate from the MMP family as well as the a disintegrin and metalloproteinase with the thrombospondin motifs (ADAMTS) family. Twenty-eight kinds of MMP have been found so far, with MMP-1, -3, and -13 playing a central role [57]. MMP-3 can degrade a broad spectrum of ECM components, including collagen, PG, laminins, and fibronectin [58]. The MMP varieties do not exist normally. Their protein synthesis can be initiated following gene transactivation by stimulation of cytkines such as IL-1 and TNF. This newly synthesized MMP remains inactivated unless a serine protease (SP) such as plasmin or trypsin is present. Our study revealed significant inhibition of PG degradation of hNSC by Decoy. Despite the significant difference between the control and IL-1 stimulation groups in the protein levels of MMP-3, there was no significant difference in PG turnover assay in this study. Several factors explain this phenomenon.

Firstly, there are influences from the other matrix-degradation enzymes which may impact the results (e.g., from other MMP and ADAMTS families). Secondly, pre-upregulation of IL-1 is possible. In one of the specimens, IL-1 may have been upregulated due to previous tissue trauma. There may be an association as to what is observed with articular cartilage, which occasionally responds poorly to IL-1 stimulation [59]. Cawston et al. [60] reported that there was no significant difference in collagen degradation in bovine nasal cartilage and human articular cartilage with or without IL-1. Thirdly, there may be limitations of our model using in vitro conditions. The concentration (10 ng/mL) of SP needs to be higher for in vitro conditions in order to activate MMP [61]. For this reason, the difference noted between the control and IL-1 group in MMP-3 protein levels did not affect PG turnover. Decoy also inhibited PG degradation in the PG turnover assay of rNSChs even in the absence of IL-1 stimulation. These results suggest that NF-κB is involved in PG metabolism (balance between synthesis and degradation) in rNSChs under normal conditions, and that Decoy inhibited NF-κB. Decoy did not suppress PG synthesis, and, with Decoy, the PG reduction rate was slower, suggesting that Decoy does not completely stop metabolism.

Although gene expression of the inflammatory markers (TNF, NO, and IL-6) and enzymes (the MMP and ADAMTS families) has been reported to be attenuated in human IVD cells by Decoy [62], we found that the protein levels of these inflammatory markers were not attenuated by Decoy. Tissue specimens from two patients showed that Decoy inhibited these inflammatory markers (Figure 8), but in the rest of the specimens, it did not. These results may vary by cell type, individual, and presence of extracellular matrix. Other transcription factors may also be involved in the regulation of IL-6 [53]. In vivo studies targeting endothelial cells in mice did not show IL-6 suppression similar to this study [53]. NF-κB is composed of at least five related proteins, p50/p105, p52 (p49)/p100, p65 (RelA), RelB, and c-Rel [63]. These subunits form a complex mixture of homodimers and heterodimers with different affinities for related, but distinct, DNA sequences [63]. The appearance of individual NF-κB protein species also varies in their responses in different cells and even in the same cell over time [63]. In addition, single-stranded Decoy, which showed similar TNF-reducing effects to double-stranded Decoy, has also been found to act on p50, but its binding is more fragile [53]. The NF-κB p50 subunit is generated from the limited proteasomal processing of the p105 precursor protein which also functions as an inhibitor of NF-κB activation [64]. p50/p50 homodimer is a transcriptional repressor, but can act as a transcriptional activator when associated with BCL3 [65]. Although Decoy clearly binds to p65 and represses its transcription, it does not appear to be regulated by a single mechanism alone.

One of the limitations of this study was that only the analyses described above could be performed due to the limited amount of tissue sample from the patients. In future work, it would be necessary to study the behavior of other inflammation markers, enzymes, and other ECMs (e.g., collagen II and hyaluronic acid) with the use of Decoy. It would be helpful to learn how Decoy interacts with different tissues (IVD and NSC, etc.), and whether any variations in the cartilage specimens would have an impact on the outcome. It would also be necessary to verify whether Decoy has the same chondroprotective effect on transplanted cartilage in vivo in animals, as shown in this study.

Glucocorticoids (GCCs) have anti-inflammatory effects by inhibiting NF-κB activation [66], but GCCs are not suitable for cartilage transplantation because of their immunosuppressive side effect. Moreover, GCCs inhibit ECM synthesis in human chondrocytes [67]. Other studies in our lab found that when dexamethasone (1 μM) was compared with Decoy (10 μM), it significantly inhibited PG synthesis in human IVD annuls fibrosus cells.

In the present study, Decoy showed no attenuation of PG synthesis and was highly effective for PG degradation, suggesting that Decoy may be useful in protecting transplanted cartilage. Theoretically, Decoy could be added to grafts during a pre-incubation period prior to their implantation, or used as an external protective material, such as a gel. In addition to the transplantation of NSC, studies have harvested this tissue for amplification as a cell source for cartilage regeneration and subsequently transplanted it into the face, nose, and joints [6,68,69]. It is for this reason that agents that support the maintenance of ECM, such as Decoy, may be useful.

## 5. Conclusions

The integrity and structural form of cartilage grafts that are used for reconstructive purposes can be compromised by degrees of graft resorption, warping from localized biomechanical forces, insufficient nutrition from recipient cells, and the inflammatory process. To curb the deleterious effect of the inflammatory process, this study was conducted to explore the potential of Decoy as a chondroprotective agent for cartilage grafts used for reconstruction. The study results demonstrate significant inhibition of PG decomposition by suppressing MMP-3 production in an hNSC explant culture system (cartilage transplantation model). Consequently, it appears that Decoy can maintain the ECM of cartilage and may be considered a novel chondroprotective agent in the cellular environment of inflammation which occurs after cartilage transplantation. Methods of application of Decoy into the wound bed are reserved for further studies.

## Figures and Tables

**Figure 1 bioengineering-11-00046-f001:**
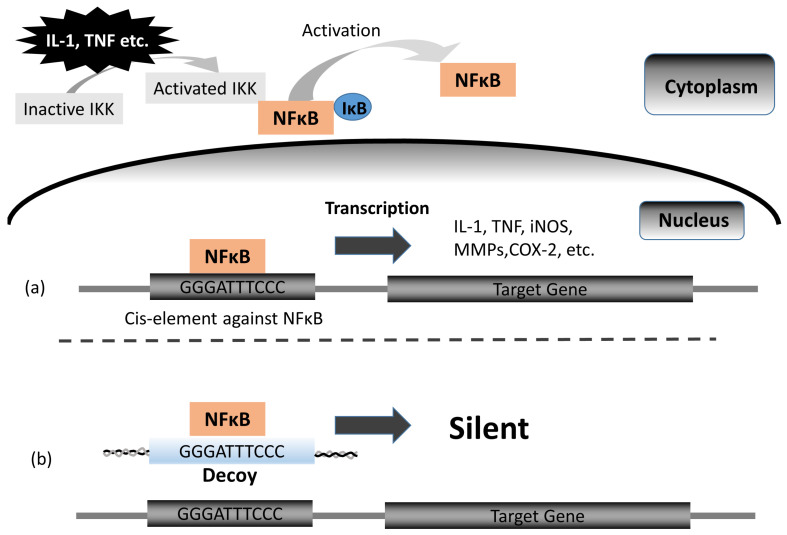
Mechanism of Decoy: (**a**) In the classical pathway, NF-κB is normally present in the cytoplasm in an inactivated state by binding inhibitor proteins including IκB. NF-κB is normally present as an inactive, Igκ-bound complex in cytoplasm. Multi-subunit IκB kinase (IKK) is activated by stimuli such as IL-1 and TNF, and IκB is inducibly degraded by activated IKK. Free NF-κB rapidly enters the nucleus and transactivates target genes such as IL-1, TNF, iNOs, MMPs, COX-2, and IL-6. (**b**) Under the presence of Decoy, transactivation will not begin because the activated NF-κB is bound to Decoy. Abbreviations: iNOs (inducible nitric oxide synthase), MMPs (matrix metalloproteinases), COX-2 (cyclooxygenase-2).

**Figure 2 bioengineering-11-00046-f002:**
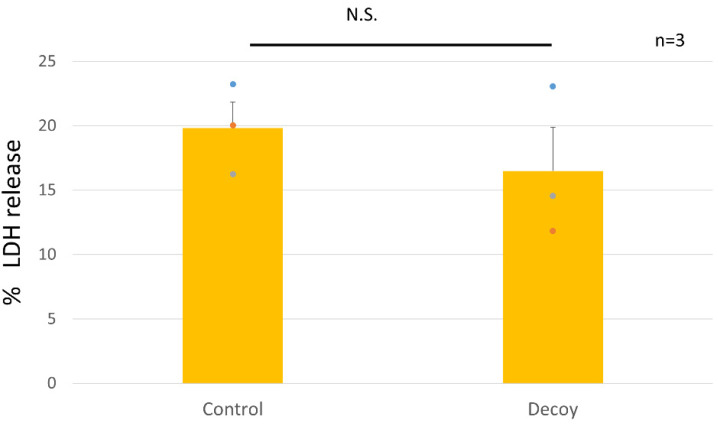
Cytotoxicity (LDH) assay of rNSChs. Cytotoxicity of Decoy was not demonstrated in the LDH assay (*p* = 0.44). The plotted values are indicated as mean ± standard error of the mean. (n = 3 biological replicates). N.S., not significant.

**Figure 3 bioengineering-11-00046-f003:**
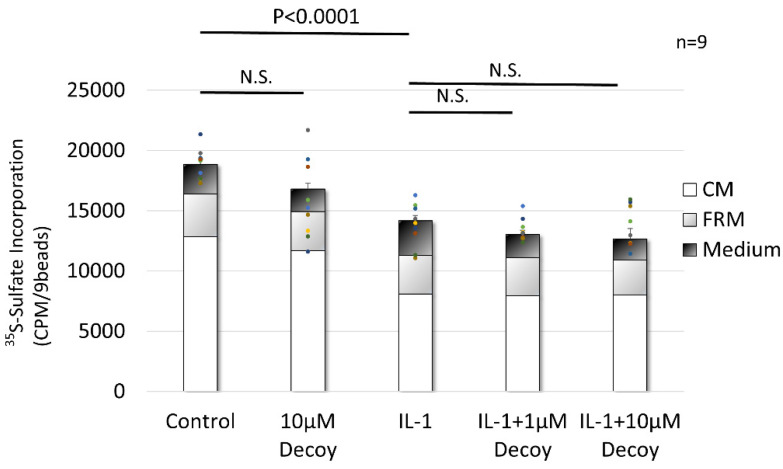
PG synthesis assay of rNSCh. IL-1 treatment significantly decreased PG synthesis (*p* < 0.0001). The 10 μM Decoy group did not inhibit PG synthesis. Decoy did not inhibit the attenuation of PG synthesis by IL-1. Data are expressed as the mean ± standard error of the mean (n = 9, 3 batches in triplicate). One-way ANOVA with Games–Howell as a post hoc test was used. N.S.; not significant.

**Figure 4 bioengineering-11-00046-f004:**
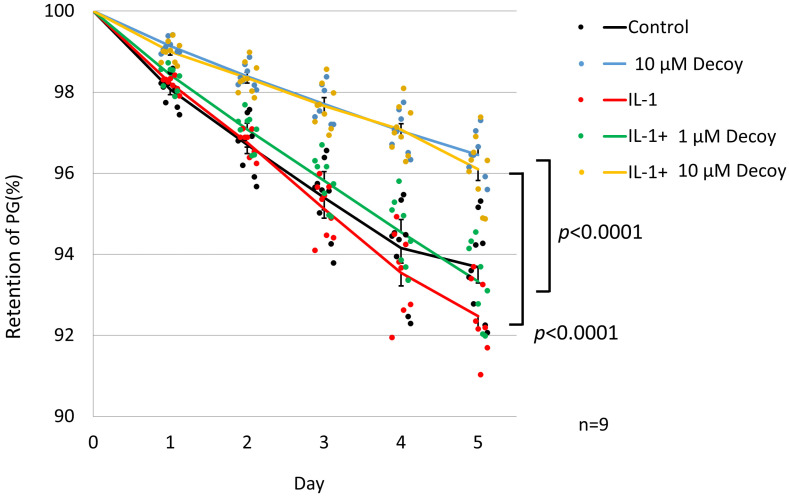
PG turnover assay of rNSCh. Dots were individual data. Compared with controls, the 10 μM decoy group showed significant inhibition of PG degradation (*p* < 0.0001). Although PG retention in the IL-1 group was lower than in the control group, there was no significant difference. In the presence of IL-1, Decoy at 1 μM did not inhibit PG degradation, while 10 µM Decoy inhibited degradation of PG significantly (*p* < 0.0001). The line in the graph shows the average for each group. Data are expressed as the mean ± standard error of the mean (n = 9, 3 batches in triplicate). Two-way repeated ANOVA with Games–Howell as a post hoc test was used.

**Figure 5 bioengineering-11-00046-f005:**
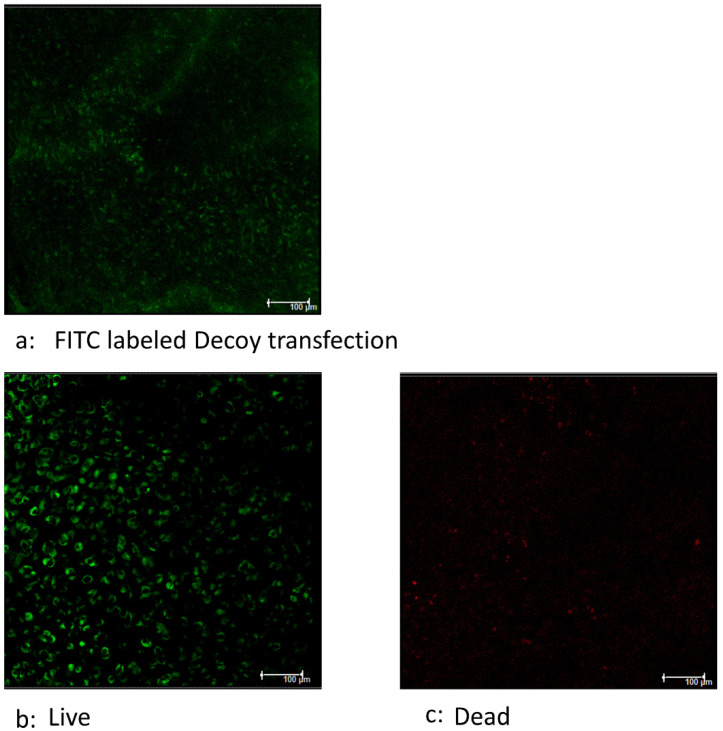
Efficiency of transfection of Decoy. Live–dead assay was separately performed using human nasal septal cartilage tissues. (**a**) Green indicates FITC-labeled Decoy 6 h after the transfection of FITC decoy. (**b**,**c**) The separate tissues were subjected to live–dead assay. Green indicates viable cells. Red indicates dead cells. Most of cells were viable in the tissues. From these images, it was inferred that Decoy was incorporated into viable cells.

**Figure 6 bioengineering-11-00046-f006:**
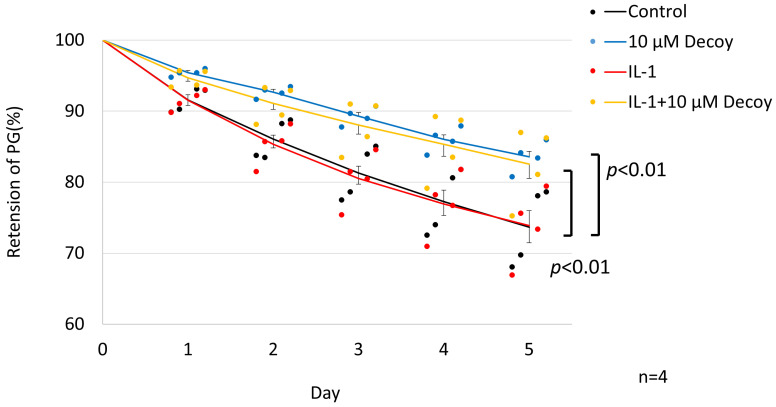
PG turnover assay of tissue culture. Dots were individual data. Decoy significantly inhibited degradation of PG with or without IL-1 (*p* < 0.01, *p* < 0.01). There was no significant difference between the control group and the IL-1 group. Data are expressed as the mean ± standard error of the mean (n = 4 patients, in triplicate). Three-way repeated ANOVA with Games–Howell as a post hoc test was used.

**Figure 7 bioengineering-11-00046-f007:**
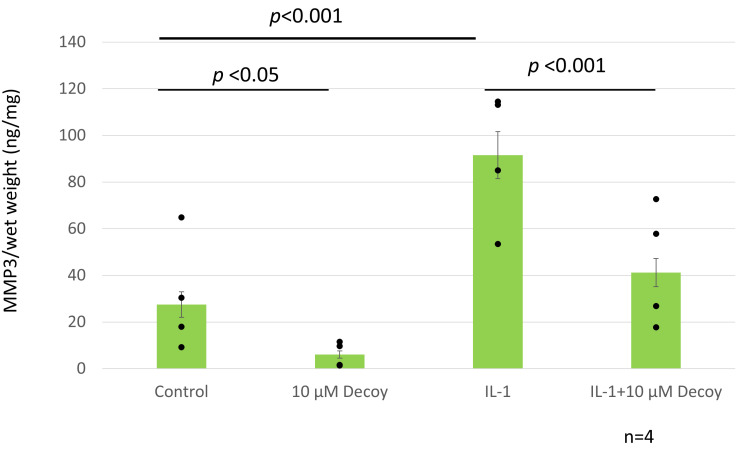
ELISA MMP-3. Dots were individual data. MMP-3 was decreased by more than 70% with administration of Decoy compared with the controls (*p* < 0.05). IL-1 treatment increased MMP-3 to 301.3% of controls (*p* < 0.001). In the presence of IL-1, MMP-3 was also significantly reduced by 51.1% with Decoy administration (*p* < 0.001). Data are expressed as the mean ± standard error of the mean (n = 4 patients, in triplicate). Two-way ANOVA with Games–Howell as a post hoc test was used.

**Figure 8 bioengineering-11-00046-f008:**
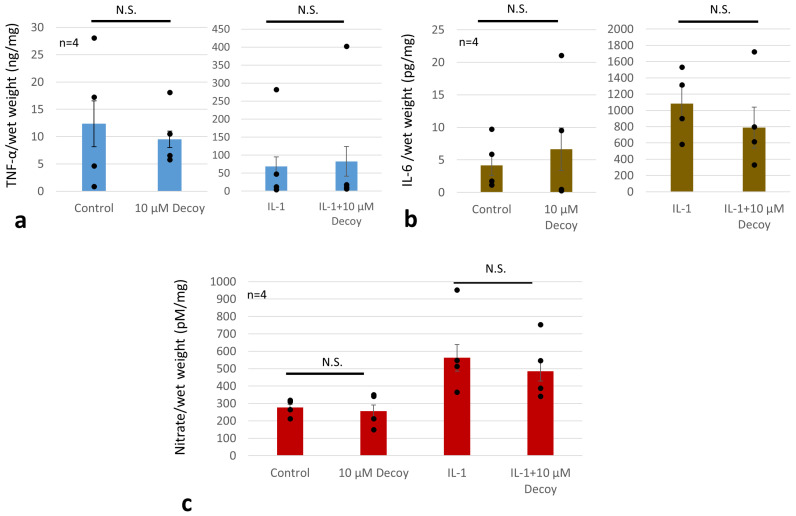
ELISA TNF, IL-6, and nitrate assay. Dots were individual data. (**a**) ELISA TNF. Decoy did not inhibit TNF. Data are expressed as the mean ± standard error of the mean (n = 4 patients, in triplicate). Two-way ANOVA with Games–Howell as a post hoc test was used. (**b**) ELISA IL-6. Decoy did not inhibit IL-6. Data are expressed as the mean ± standard error of the mean (n = 4 patients, in triplicate). Two-way ANOVA with Games–Howell as a post hoc test was used. (**c**) Nitrate assay. Decoy did not inhibit nitrate. Data are expressed as the mean ± standard error of the mean (n = 4 patients, in triplicate). Two-way ANOVA with Fisher’s LSD as a post hoc test was used. N.S.; not significant.

## Data Availability

Data is contained within the article.

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
