# Peer review of "Nuclear Factor-κB Decoy Oligodeoxynucleotide Attenuates Cartilage Resorption In Vitro"

_bioengineering, 2024, doi:10.3390/bioengineering11010046_

Round 1

Reviewer 1 Report (Previous Reviewer 1)

Comments and Suggestions for Authors

Comments and Suggestions for Authors:

I have received the manuscript "Nuclear Factor-kB Decoy oligodeoxynucleotide attenuates cartilage resorption in vitro" by Nemato et al. for review three times. The work is very interesting. After the first time, I described in long detail what needs to be modified to make it suitable for publication. I have compared the third version with the first and unfortunately found only minimal differences and therefore I recommend rejection of the manuscript.

Author Response

We would like to thank the reviewers for their time and effort in reading our work and for providing valuable suggestions to improve the manuscript; changes, wherever possible, have been incorporated into our revised version. Our responses to the reviewers’ list of questions are described below:

Reviewer 1:

I have received the manuscript "Nuclear Factor-kB Decoy oligodeoxynucleotide attenuates cartilage resorption in vitro" by Nemato et al. for review three times. The work is very interesting. After the first time, I described in long detail what needs to be modified to make it suitable for publication. I have compared the third version with the first and unfortunately found only minimal differences and therefore I recommend rejection of the manuscript.

We appreciate the sequential reviews from R1, and have provided earnest responses and thoughtful corrections to the points raised. Regarding the status of the references older than 30 years, we needed to retain the “original sources” because they fall into the category of “groundbreaking methodological publications.” However, they account for only 13/65 of the references, or 20%. Recent literature of less than 5 years is also included in the manuscript and represent 16/65 of all the references, which is 24.6% of the total.

We are grateful to the reviewers for their time and effort in our manuscript submission.

Sincerely yours,

Dr. Hitoshi Nemoto

Dr. Daisuke Sakai

Dr. Deborah Watson

Dr. Koichi Masuda

Reviewer 2 Report (New Reviewer)

Comments and Suggestions for Authors

The authors investigated the effect of NFκB decoy oligodeoxynucleotide on cartilage resorption by assessing IL-1 induced PG degradation and turn over, MMP3, IL-6, NO, and TNFa production in rabbit nasal septal chondrocyte and human nasal septal cartilage. Referring published data, the authors assumed these enzymes and cytokines are regulated by proinflammatory cytokine induced NFkB signaling. Although a significant downregulation of PG and MMP3 by the Decoy has been observed, IL-6, TNFa, and NO secretion have not been affected by the Decoy in their isolated human nasal septal cartilage tissues. Based on the in vitro data, the authors suggested the NFκB decoy could be considered as a novel chondroprotective agent after cartilage transplantation.

Major concerns:

1.       This reviewer would suggest to try some histochemistry or cytochemistry analysis of collagen I or II. or hyaluronate in IL-1 or TNFa stimulated chondrocyte, to “composite” non-regulated IL-6/TNFa/NO.

2.       IT would be better to address how long the incorporated Decoy can remain in transfected chondrocytes.

Minor concerns:

1.       Please reword line 15-16: “Transcriptional factor nuclear factor κB (NFκB) is activated by cytokines, such as interleukin-1 (IL-1)…”. It sounds like all cytokines activate NFkB. Many cytokines engage their cognate receptors and signal through STATS but not NFkB. Such as IL4, IL-6, IFNs…

2.       Line 58, “the NF-κB binding site (GGGATTTCCC)…” which NFkB? According to the sequence, it looks like the classical NFkB-P65 binding…

3.       In line 160, 161: “Decoy (0, 10, 0,1and 10 μM)..” it is hard to understand.

4.       Line 265, what is “ODN”?

5.       Most of content in Line 306-328 could be moved to “Introduction”,

6.       The quality of figures could be improved.

Author Response

We would like to thank the reviewers for their time and effort in reading our work and for providing valuable suggestions to improve the manuscript; changes, wherever possible, have been incorporated into our revised version. Our responses to the reviewers’ list of questions are described below:

Reviewer 2:

Major concerns:

  1. This reviewer would suggest to try some histochemistry or cytochemistry analysis of collagen I or II or hyaluronate in IL-1 or TNFa stimulated chondrocyte, to “composite” non-regulated IL-6/TNFa/NO.

This is an excellent suggestion but the specimens were exhausted when running them through the assays. Your suggestion is valuable feedback and we have added this as a limitation of our study within the Discussion.

  1. IT would be better to address how long the incorporated Decoy can remain in transfected chondrocytes.

Although Decoy is known to disappear rapidly, it is currently unclear how long it actually remains in the chondrocytes. To be more complete, we have added a description of the pharmacokinetics of ODN therapy into the manuscript.

Minor concerns:

  1. Please reword line 15-16: “Transcriptional factor nuclear factor κB (NFκB) is activated by cytokines, such as interleukin-1 (IL-1)…”. It sounds like all cytokines activate NFkB. Many cytokines engage their cognate receptors and signal through STATS but not NFkB. Such as IL4, IL-6, IFNs…

Thank you for pointing this out. The following has been added to the manuscript in the Background: “There are two types of NFκB activity pathways (classical and non-classical): the classical pathway is activated by proinflammatory cytokines such as TNF, IL-1, IL-6, chemokines, and anti-microbial proteins cytokines, while the non-classical pathway is activated by CD40L and B cell activating factor (BAFF)”.

  1. Line 58, “the NF-κB binding site (GGGATTTCCC)…” which NFkB? According to the sequence, it looks like the classical NFkB-P65 binding…

You are correct regarding the classical NFkB-P65 binding site and this is now mentioned in the text.

  1. In line 160, 161: “Decoy (0, 10, 0,1and 10 μM)..” it is hard to understand.

Thank you for noting the typo; the lines have been altered for clarity.

  1. Line 265, what is “ODN”?

ODN is the abbreviation for oligodeoxynucleotide; "ODN" was deleted because NFκB decoy oligodeoxynucleotide was abbreviated to “Decoy”.

  1. Most of content in Line 306-328 could be moved to “Introduction”.

This suggestion to shift this content to the Introduction has been completed.

  1. The quality of figures could be improved.

The quality of the figures have been replaced with higher resolution images.

We are grateful to the reviewers for their time and effort in our manuscript submission.

Sincerely yours,

Dr. Hitoshi Nemoto

Dr. Daisuke Sakai

Dr. Deborah Watson

Dr. Koichi Masuda

Reviewer 3 Report (New Reviewer)

Comments and Suggestions for Authors

Thank you very much for submitting your interesting manuscript to Bioengineering. As a reviewer, I have the following comments on this study:

[Major comments]

1. In vivo studies should be displayed. In this case, the authors should include a positive control such as emicizumab or filgotinib. Additionally, mechanistic studies by the authors or other scientists should be displayed in the Discussion section.

2. Please add some positive controls in Figs. 3~8. These should be clearly performed and explained.

[Minor comments]
-Please unify the description of the words such as nuclear factors, cytokines, and proteins. NF-kB or NFkB, MMP-3 or MMP3, 10 uM or 10uM etc. There are so many typos in the main text. Please correct them.
-The novelty of this paper is not clear. The Introduction section as well as the Discussion section are poorly described. Strengthen the sections.

-In the Materials and Methods section, why do the authors divide the subtitle? Simplify to unify them.

Comments on the Quality of English Language

There are so many typos in the main text. 

Author Response

We would like to thank the reviewers for their time and effort in reading our work and for providing valuable suggestions to improve the manuscript; changes, wherever possible, have been incorporated into our revised version. Our responses to the reviewers’ list of questions are described below:

Reviewer 3:

  1. In vivo studies should be displayed. In this case, the authors should include a positive control such as emicizumab or filgotinib. Additionally, mechanistic studies by the authors or other scientists should be displayed in the Discussion section.

We appreciate this valuable feedback; however, without the same patient specimens, we are no longer able to run in vivo studies. The pharmacokinetics of ODN, including Decoy, was described in the Discussion section.

  1. Please add some positive controls in Figs. 3~8. These should be clearly performed and explained.

Similar to the response to comment 1, the patient specimens used in this study have been exhausted from use in the assays; additional testing is no longer possible.

[Minor comments]

-Please unify the description of the words such as nuclear factors, cytokines, and proteins. NF-kB or NFkB, MMP-3 or MMP3, 10 uM or 10uM etc. There are so many typos in the main text. Please correct them.

-The novelty of this paper is not clear. The Introduction section as well as the Discussion section are poorly described. Strengthen the sections.

We appreciate your detailed review of our manuscript and have corrected the typos and unified the word descriptions.

-In the Materials and Methods section, why do the authors divide the subtitle? Simplify to unify them.

We have reduced the number of subtitles in the Materials and Methods section and simplified it. Similarly, the results section has been altered in the same manner.

We are grateful to the reviewers for their time and effort in our manuscript submission.

Sincerely yours,

Dr. Hitoshi Nemoto

Dr. Daisuke Sakai

Dr. Deborah Watson

Dr. Koichi Masuda

Round 2

Reviewer 3 Report (New Reviewer)

Comments and Suggestions for Authors

Thank you very much for correction. But in some response, there should be added further description and/or data. Please read carefully and respond them. 

Comments on the Quality of English Language

Minor correction is needed.

Author Response

We appreciate the reviewers time and effort in reading our work and for providing valuable suggestions to improve the manuscript. Changes, wherever possible, have been incorporated into our revised version. Our response to the comment from Reviewer 3 is entered below:

Reviewer 3:

Thank you very much for correction. But in some response, there should be added further description and/or data. Please read carefully and respond them.

We have included additional information into the Background section to further emphasize the meaning, significance, and necessity of this study. We have also added the diversity of Decoy responses by cell type in the Discussion section in conjunction with other mechanistic studies. As mentioned previously, repeating any of the studies would not be possible because the samples have been exhausted.

We are grateful to the reviewers for their time and effort in our manuscript submission.

Sincerely yours,

Dr. Hitoshi Nemoto

Dr. Daisuke Sakai

Dr. Deborah Watson

Dr. Koichi Masuda

This manuscript is a resubmission of an earlier submission. The following is a list of the peer review reports and author responses from that submission.

Round 1

Reviewer 1 Report

Comments and Suggestions for Authors

Comments and Suggestions for Authors:

Nemoto and co-workers describe the effectiveness of Decoy oligodeoxynucleotide in inhibiting cartilage degradation. The work is very interesting and deserves publication, but not in this form. I recommend that the paper be rejected in its current form, but encourage the authors to rewrite and resubmit the revised manuscript.

My suggestions are as follows:

1. The introduction and the discussion should be written in a way that includes the latest research findings. In general, 50% of the references should be from the last 5 years. Out of 52 references in your manuscript, only 3 are from the last 5 years, if I counted correctly.

2. The TNF is no longer classified as "alpha" due to the reclassification of TNF beta as lymphotoxin - please correct it throughout.

3. The explanation of the Decoy mechanism is not clear. (see lines 58-59).

4. Figure 1 does not explain what IKK, IKKβ is.

5. Figure 1b is not correct. I understand the authors' idea that there is no gene expression when the transcription factor is linked to the decoys, but the scheme should be changed. The decoys cannot be silent only the gene.

6. The reference number of the material is missing everywhere in the materials and methods. (Example, it should be like this: primary rabbit anti-CD163 (ab182422, Abcam).

7. The results are interesting, but the description and naming of them are not good. You cannot describe part 1 or part 2.

8. Each experiment should consist of a new paragraph with its own name. Example: The decoy oligonucleotide effectively inhibited the release of LDH.

9. The reader should be helped to read the results. Why we are doing the experiment, what exactly we are doing, and what the conclusion is.

10. One paragraph should be linked to the other. We've seen this in the previous paragraphs, so we'll go ahead and look at this…

11. In Figure 5, the size of the scale bars is not centered.

12. Figure 5 is not of good quality.

13. The discussion should be supplemented with new information from the last 5 years.

Reviewer 2 Report

Comments and Suggestions for Authors

This is a well-written and well-presented study with remarkable results. I have no comments to improve the paper and suggest it to be published in the current format. But before publication please add the authors' contributions to the MS.

Reviewer 3 Report

Comments and Suggestions for Authors

In the manuscript, Nemoto and colleges investigate the potential of decoy oligonucleotide on the implanted cartilage resorption by ameliorating NF-kb signaling activation. The study is intuitive and interesting. Several questions are raised for authors to improve the manuscript.

1.      IL-6 inhibits PG synthesis in rNSCh cells. Does IL-1 inhibit PG synthesis in hNSC cells? Could authors determine the effects of their treatment on the expression of PG synthesis genes?

2.      In both rNSCh and hNSC Decoy oligonucleotide could inhibit PG degradation independent from the IL-1 treatment, could the authors make some discussion on these results?

3.      The authors do not employ control oligodeoxynucleotide in their study, please revised.

4.      Since multiple inflammatory cytokines, including IL-1 and TNFa, transmit signals through the NF-kb pathway, could authors determine the effective dose of decoy to turn down the NF-kb signaling and the effect on PG degradation when multiple cytokines are present simultaneously at physiological concentration?

5.      Decoy oligonucleotide downregulates inflammatory signaling by competing for the binding of a transcription factor with promoters of responsive genes, could authors measure the RNA transcription to make the study more comprehensive?

6.      Could authors monitor the half-life of the fluorescence-labeled Decoy oligonucleotide within cells? Does it last days?

Round 2

Reviewer 1 Report

Comments and Suggestions for Authors

I congratulate the authors because the manuscript has improved a lot.

I am very sorry, however, but the manuscript is still not up to the standard of the "Bioengineering" journal. Among my suggestions, I do not see any significant changes in 1,9,10,11,12.

Reviewer 3 Report

Comments and Suggestions for Authors

Dear Authors,

  After carefully reviewing your revised material, the reviewer is regretting to receive inadequate responses to the raised concerns for your study. Since these concerns are critical for the results of the study, with no further clarification, I could not support the publication of the current manuscript at this moment.